# Pediatric Functional Abdominal Pain Disorders following COVID-19

**DOI:** 10.3390/life12040509

**Published:** 2022-03-30

**Authors:** Mioara Desdemona Stepan, Ramona Cioboata, Ştefăniţa Bianca Vintilescu, Corina Maria Vasile, Andrei Osman, Mircea Sorin Ciolofan, Mihaela Popescu, Ilaria Lorena Petrovici, Andrei Calin Zavate

**Affiliations:** 1Department of Infant Care-Pediatrics-Neonatology, University of Medicine and Pharmacy of Craiova, 200349 Craiova, Romania; dstepan80@yahoo.com; 2Department of Pneumology, University of Pharmacy and Medicine Craiova, 200349 Craiova, Romania; ramona.cioboata@umfcv.ro; 3Department of Pediatric Cardiology, University of Medicine and Pharmacy of Craiova, 200349 Craiova, Romania; 4Department of Anatomy and Embryology, University of Medicine and Pharmacy of Craiova, 200349 Craiova, Romania; andreiosman3@gmail.com; 5Department of ENT, University of Medicine and Pharmacy of Craiova, 200349 Craiova, Romania; sorin.ciolofan@yahoo.com; 6Department of Endocrinology, University of Medicine and Pharmacy of Craiova, 200349 Craiova, Romania; mihaela.n.popescu99@gmail.com; 7Department of Pediatric Surgery, University of Medicine and Pharmacy of Craiova, 200349 Craiova, Romania; ilaria.petrovici@yahoo.com (I.L.P.); azavate@gmail.com (A.C.Z.)

**Keywords:** COVID-19, pandemic, children, functional abdominal pain disorders

## Abstract

Background: Functional abdominal pain disorders (FAPD) are a group of functional gastrointestinal disorders with multifactorial etiology and are subclassified using Rome IV criteria into a series of clinically distinct entities represented by irritable bowel syndrome, functional dyspepsia, abdominal migraine and functional abdominal pain that is not otherwise specified. Digestive functional disorders associated with severe acute respiratory syndrome coronavirus-2 (SARS-CoV-2) may be mediated by the involvement of complex pathogenic mechanisms, which have been under investigation in children since the beginning of the coronavirus disease pandemic (COVID-19). Methods: In this retrospective, observational descriptive and analytical study, we investigated the presence of chronical functional abdominal pain in preschool children (4–6 years old) from the south-west of Romania in the pre-pandemic (18 cases) and COVID-19 pandemic period (34 cases), as well as the association with the COVID-19 positive and COVID-19 negative statuses, gender, environment origin, and viral infection-associated symptoms. Age-specific Rome IV criteria were used to diagnose functional abdominal pain. We performed an integrated statistical analysis of the results utilizing an electronic database in which we compared the data in order to assess the impact of COVID-19 on the clinical-epidemiological parameters analyzed. Results: In the pre-pandemic group, irritable bowel syndrome predominated (77.8%), followed by functional dyspepsia (22.2%), the other types of functional abdominal pain being absent, while for the pandemic group, irritable bowel syndrome was the most common (79.4%), followed by abdominal migraine (11.8%), abdominal dyspepsia (5.9%) and functional abdominal pain not otherwise specified (2.9%). We found a female/male ratio difference of 0.84 and an urban/rural ratio of 1.83 in favor of the pandemic group. These discrepancies were mainly caused by the differences between the COVID-19 positive and negative pandemic groups, where we observed statistical association of the positive pandemic group with IBS and urban environment, and a tendency of FAPDs diagnostic mainly with males. The predominant symptoms associated with COVID-19 positive cases were digestive (60.9%) or respiratory (39.1%). Conclusions: Our study demonstrates viral-mediated sensitivity of the gastrointestinal tract in preschool children, considering different clinical-epidemiological profiles related to the prevalence of FAPD and according to gender and environment origin, while the contribution of the pandemic context remains to be demonstrated in larger studies

## 1. Introduction

Coronavirus disease 2019 (COVID-19) is caused by a newly identified coronavirus, named severe acute respiratory syndrome coronavirus 2 (SARS-CoV-2), which belongs to the family Coronaviridae, subfamily Orthocoronavirinae, a virus exhibiting the largest positive-sense single-stranded RNA genome [1,2,3]. The first cases of patients affected by the new coronavirus were documented in December 2019 within China’s Hubei Province, Wuhan City, with the initial outbreak spreading so rapidly that in March 2020 it was declared a pandemic by the World Health Organization [1,2,4].

In COVID-19 positive patients, the symptoms differ strikingly from asymptomatic to mild, severe respiratory symptoms, multiorgan failure, and even death [5,6,7]. Although respiratory manifestations are often present or dominant, an increased rate of cases with somatic or functional digestive symptoms with concomitant viral or postinfectious manifestations has been found in children [4,5,8]. Therefore, while in adults the gastrointestinal symptoms induced by viral infection reach up to 61%, among the pediatric population they can be present in up to 84.1%, with abdominal pain being one of the common digestive symptoms of SARS-CoV-2 infection in children [5]. In this context, abdominal pain in functional gastrointestinal disorders (FGID) should be differentiated from the one present in Pediatric Multisystem Inflammatory Syndrome temporally associated with COVID-19 which may mimic a condition requiring surgery or may mask cardiac or multiorgan damage that can sometimes be fatal [5,8]. FGIDs affect 5–20% of the general population and have a multifactorial etiology, including a broader group of clinical entities represented by functional abdominal pain disorders (FAPD), along with vomiting, aerophagia, constipation, and incontinence [4,9]. In turn, FAPD shows a prevalence of 1.6–41.2% based on country origin, age, and sex, affecting up to 25% of the worldwide pediatric population [10,11,12]. According to the Rome IV diagnostic criteria, FAPD can only be identified after appropriate medical evaluation and the symptoms cannot be attributed to another condition [4]. Functional dyspepsia (FD), irritable bowel syndrome (IBS), abdominal migraine (AM) and functional abdominal pain not otherwise specified (FAP-NOS) are the four conditions that are included in the notion of FAPD [13,14,15,16,17,18]. Changes that occur during COVID-19 may underlie the initiation and progression of FAPD through etiopathogenic mechanisms that are still unclear, which equally apply to viral and digestive disease. Intestinal, central, and extrinsic infectious factors are involved in the etiopathogenesis of FAPD damaging the microbiota-intestine-brain axis [10,13]. Loss or reduction of intestinal absorption, mucosal inflammation, abnormal function of ACE-2 (angiotensin-converting enzyme-2) receptors, and psychosocial and microbiota changes in the digestive tract during COVID-19 are possible explanations for the onset or exacerbation of functional gastrointestinal disorders in preschoolers [13,15,19].

Due to its incidence, the difficulty of positive and differential diagnosis, as well as the management complexity, abdominal pain is a unique pattern when investigating functional disorders in children in general. There are relatively few studies in the literature regarding the distribution of FAPD in the pediatric population in the context of SARS-CoV-2 infection, and the results are controversial and difficult to interpret due to the heterogeneity of the groups analyzed and the diagnostic and investigative methods. They are also influenced by the criteria for the diagnosis of FAPD, which were not always represented by the Rome IV criteria. Finally, many studies rely on physical or online questionnaires to assess diagnostic criteria [4,20].

In this context, our study aims to investigate FAPD in preschool children affected by viral infection, according to the latest diagnostic criteria, which may contribute to determining the role of COVID-19 in the occurrence of these clinical manifestations.

## 2. Materials and Methods

### 2.1. Study Setting and Study Design

In this retrospective, observational, descriptive study with an analytical component we studied the presence of chronically manifested FAPD in the pediatric population in the south-western of Romania, compared to the pre-pandemic and pandemic COVID-19 period, to highlight the impact of viral infection on digestive manifestations. The potential viral involvement in digestive disorders was analyzed based on the clinical-epidemiological characteristics of the patients.

### 2.2. Study Population and Inclusion Criteria

In this study, two groups of preschool children aged between 4 and 6 years were included. Therefore, the first group considered pre-pandemic (not exposed to SARS-CoV-2 infection), included patients who were presented between 1 February 2019–1 August 2019 in the Pediatric Department of the Craiova Emergency Clinical County Hospital for chronic abdominal pain; and this group is further on considered as the control group. The second group considered pandemic (exposed to SARS-CoV-2 infection) included patients who presented between 1 February 2021–1 August 2021 in the same department and with the same symptoms. The inclusion criterion in the study was the presence of the diagnosis represented by FAPD, which was made using Rome IV criteria applicable for preschool age 4–6 years [21]. Within the pandemic group (exposed to SARS-CoV-2 infection), we considered the existence of two different subgroups represented by the COVID-19 negative pandemic group and the COVID-19 positive pandemic group.

### 2.3. Sample Size and Sampling Methods

The sampling technique was represented by the non-probability quota sampling method, in which we selected all patients from the time intervals of interest who fulfilled the inclusion and exclusion criteria. All these criteria aimed to ensure homogeneity and representativeness of the study groups. The pre-pandemic group included a total of 18 patients and the pandemic group consisted of 34 patients. All children were at least 4 years old and not more than 6 years old for being classified as preschoolers, and for the Rome IV criteria to be applicable for this age range to which the study addressed to. For children under 4 years old, the functional pain has a different clinical profile and implies different diagnostic criteria than those used by us. We also considered that the transition to school-age implies behavioral, social, nutritional, and intestinal microbiota changes with a major influence on functional abdominal pain.

In order to reduce environmental variables as much as possible, both the pandemic and pre-pandemic groups were selected from the same period of the year. For both groups, only patients who had no infections (including SARS-CoV-2) or chronic inflammation in the last 3 months were included, thus ruling out the possibility of receiving prior specific medical or surgical treatments; also, only patients without immunocompromised statuses were included, as this might have an important impact on the perception of abdominal pain, through the effect on the gastrointestinal microbiota.

### 2.4. Exposure Assessment

In our study the exposure assessment for SARS-CoV-2 infection was given by the positivity or negativity of RT-PCR tests, but the exclusion or inclusion in the positive or negative pandemic groups took into account both the viral status at the time of the medical presentation in which the FAPD diagnosis was established, as well as the results of the RT-PCR tests before or after this medical visit. Due to the persistent chronic clinical profile that defined FAPDs, all patients selected in the pandemic group for this study had previous RT-PCR tests. The inclusion of patients in the positive pandemic group was made on the basis of positive RT-PCR tests performed at most 6 months and at least 3 months before the medical presentation in which the diagnosis of FAPD was established. This was due, first, because the chronic functional pain according to Rome IV criteria persists for 2–6 months depending on the type of FAPD, and second, because we considered from our own professional experience that at more than 6 months after infection, besides the attenuation of symptoms there are difficulties in evoking, describing and perceiving functional pain associated with COVID-19. Also, the 3-month limit for a positive RT-PCR test was necessary for a patient to be able to get through the infectious period and have enough time to develop FAPD symptoms that would be present at least 2 months until the medical presentation, in which the diagnosis of FAPD is established.

In all cases, RT-PCR tests were performed at the medical presentation. All patients who were included in the study in the pandemic group were negative for RT-PCR at the medical presentation in which the diagnosis of FAPD was established, a diagnosis that cannot be made objectively at an acute infectious moment. In the context in which there may be gaps between the existence of SARS-CoV-2 symptoms and RT-PCR positivity, which are not always detected simultaneously, the viral status of patients who met the criteria for inclusion in pandemic groups at the time of medical presentation and FAPD diagnosis was also monitored.

Thereby, the patients from the positive pandemic group had positive RT-PCR tests 3–6 months before the time of medical presentation in which FAPD was diagnosed and negative RT-PCR test at presentation and after FAPD diagnosis. The patients from the negative pandemic group had only negative RT-PCR tests, including at the moment of medical presentation and after FAPD diagnosis. Simultaneously, for the pandemic group, we observed the predominance of respiratory or digestive symptoms.

### 2.5. Outcome Assessment

Functional gastrointestinal disorders (FGID) represent a wide group of pediatric functional digestive disorders with variable pathogenesis and diagnostic criteria depending on age [4]. Within this category, a particular group is represented by FAPD, diagnosed based on the Rome IV criteria developed in 2016 and specific to ages greater than or equal to 4 years [21]. These criteria separately and objectively define four-component clinical entities of FAPD, represented by irritable bowel syndrome (IBS), functional dyspepsia (FD), abdominal migraine (AM), and functional abdominal pain not otherwise specified (FAP-NOS). These criteria were assessed by a rigorous clinical-anamnestic investigation that followed the duration and time intervals for the onset of pain and associated digestive symptoms, which allowed classification of the digestive disease into one of the subtypes of FAPD (Table 1). In controversial cases, the examination was repeated, with separate or joint participation of the patients’ parents or relatives, and the results were interpreted by two specialists who reached a consensus.

### 2.6. Study Covariants

In this study, we analyzed the distribution of FAPD in the pre-pandemic and pandemic groups, as well as their distribution in relation to gender, environment origin, and symptoms associated with the viral infection. Comparisons were also made for the COVID-19 positive and COVID-19 negative pandemic groups, and for the pre-pandemic and COVID-19 negative pandemic groups.

The results were expressed descriptively as a number of cases, prevalence (%), and percentage variations for the categories analyzed. All patient-related information was tabulated into an electronic database, using Microsoft Office Excel 2010. Statistical analysis was performed using the comparison chi-squared test (χ^2^) within SPSS10 (Statistical Package for Social Sciences) software, and the value *p* < 0.05 was considered significant.

### 2.7. Ethical Principles

In this study, a written informed consent was obtained from all legal representatives of the patients regarding the processing of data for scientific purposes, the study being conducted according to the guidelines of the Declaration of Helsinki and approved by the Local Ethics Committee.

## 3. Results

General group. In this study, based on the total number of cases (52 cases), we have found that the most frequent FAPD subtype was IBS with 41 cases (78.8%), followed by FD with six cases (11.5%), AM with 4 cases (7.7%) and FAP-NOS observed in only one case (2%) (Table 1). The majority of FAPD patients were females, i.e., 33 cases (63.5%), against males with 19 cases (36.5%), with a ratio of 1.73 between the two categories. At the same time, most patients were from urban areas, as observed in 37 cases (71.2%) compared to rural areas present in 15 cases (28.8%), the urban/rural ratio being 2.46 (Table 1). For the whole group analyzed, COVID-19 symptoms were absent in 29 cases (55.8%), patients that belonged to the pre-pandemic group included 18 cases (34.6%), and to the COVID-19 negative pandemic group in 11 cases (21.2%), while COVID-19 symptoms were present in all cases of the COVID-19 positive pandemic group, i.e., 23 cases (44.2%). For the entire group, the female gender predominated in all forms of FAPD except FAP-NOS, while environment origin was variable, with urban environment prevailing in IBS and AM (Figure 1A).

In the pre-pandemic group, which included 18 cases out of the total number of patients (34.6%), IBS predominated as FAPD subtype with 14 cases (77.8%), followed by FD with four cases (22.2%), AM and FAP-NOS subtypes being absent (Table 2, Figure 1B). Also, in this group females were slightly predominant, with 10 cases (55.6%), compared to males with eight cases (44.4%) and a female/male ratio of 1.25, as well as urban with 11 cases (61.1%) compared to rural with seven cases (3.9%) and an urban/rural ratio of 1.42 (Table 2, Figure 1C,D).

In the pre-pandemic group, we noticed the association of IBS with the urban environment and FD with the female gender (Figure 2A).

In the pandemic group including 34 cases out of the total number of patients (65.4%), we found the absence of viral infection in 11 cases (32.4%) and its presence in 23 cases (67.6%) (Table 2). Reporting to the total number of cases in this group, the predominant FAPD subtype was IBS with 27 cases (79.4%), followed by AM with four cases (11.8%), FD with two cases (5.9%), and FAP-NOS observed in one case (2.9%) (Table 2, Figure 1B). Female gender predominated in this group with 23 cases (67.6%) compared to males with 11 cases (32.4%) and female/male ratio 2.09, as well as urban with 26 cases (76.5%) compared to rural with eight cases (23.5%) with an urban/rural ratio of 3.25 (Table 2, Figure 1C,D). Within the pandemic group, we found the association of IBS with urban areas and AM with female gender (Figure 2B).

Analysis of the results for the pre-pandemic and pandemic groups, in addition to the predominance of IBS as a subtype of FAPD in both groups, showed different clinical-epidemiological profiles, with the absence of AM and FAP-NOS and a double incidence of FD in the pre-pandemic group compared to the pandemic group. At the same time, we have found a difference in the female/male ratio of 0.84 and urban/rural ratio of 1.83 in favor of the pandemic group.

However, the statistical analysis of the cases distribution in the pre-pandemic and pandemic groups indicated nonsignificant differences in FAPDs (*p* = 0.146, χ^2^ test) and gender (*p* = 0.389, χ^2^ test) and environment origin. (*p* = 0.246, χ^2^ test). On the contrary, when comparing the pre-pandemic, pandemic COVID-19 negative and pandemic CO-VID-19 positive groups, we found significant associations of IBS (*p* = 0.045, χ^2^ test) and the urban environment (*p* = 0.011, χ^2^ test) with the latter. 

These differences were mainly due to differences within the pandemic group between the COVID-19 positive and negative categories. Within the COVID-19 positive pandemic group, IBS was identified in 21 cases (91.3%) and AM in two cases (8.7%), while within the COVID-19 negative pandemic group, IBS cases were present in only six cases (54.5%), followed by FD and AM with two cases (18.2%) and FAP-NOS in one case (9.1%) (Table 2, Figure 1B). The different FAPD profile between the two groups points to a variation of 28.8% for the presence of IBS in favor of the COVID-19 positive group in relation to the total number of cases in the study. We have found here a statistically significant association of the COVID-19 positive pandemic group with IBS (*p* = 0.044, χ^2^ test). Similar differences were also found for gender and environment origin parameters. Thus, for the COVID-19 positive pandemic group the number of females was of 14 (60.9%) compared to 9 males (39.1%), and with a female/male ratio of 1.55, while for the COVID-19 negative pandemic group the values were 9 (81.8%) vs. 2 (18. 2%) and a female/male ratio of 4.5, suggesting a tendency of association of viral infection with males in the COVID-19 positive group, although this group provided a considerable number of females, and has an impact when comparing the pandemic and pre-pandemic groups (Table 2, Figure 1C). However, the association of the males with the COVID-19 positive pandemic group remained only as a tendency, as no gender differences were statistically identified in the two groups (*p* = 0.222, χ^2^ test).

Finally, for the COVID-19 positive pandemic group, the number of cases in urban areas was of 21 (91.3%), while in rural areas it was of 2 (8.7%) and with an urban/rural ratio of 10.5, compared to the COVID-19 negative pandemic group where the values were 5 (45.5%) vs. 6 (54.5%) and with a ratio of 0.83 (Table 1, Figure 1D). The prevalence variation of 30.8% for the urban environment in relation to the total number of cases considered in the study was in favor of the COVID-19 positive group. The association strength of the urban environment with COVID-19 positive pandemic group was statistically confirmed (*p* = 0.003, χ^2^ test). Moreover, symptoms of viral infection were mainly represented by digestive manifestation in 14 cases (60.9%) and respiratory issues in 9 cases (39.1%) (Figure 2C).

Although IBS and female gender predominated for all groups analyzed, it is possible to state that the COVID-19 positive pandemic group had a tendency of association with cases of males diagnosed with IBS and with the urban origin, a result that is also evident when comparing the two COVID negative and positive pandemic groups (Figure 2C,D), characteristics that can be attributed to the viral etiology.

Another important comparison between the pre-pandemic and the COVID-19 negative pandemic groups has shown differences, with at least double the number of IBS and FD cases and a difference in the female/male ratio of 3.25 in favor of the latter group, which can be attributed to the pandemic situation. However, the statistical analysis indicated nonsignificant associations of case distribution in relation to FAPDs (*p* = 0.138, χ^2^ test), gender (*p* = 0.149, χ^2^ test) and environment origin (*p* = 0.411, χ^2^ test).

## 4. Discussion

FGID is a condition of stress-mediated pathology, which involves, exposure of the digestive tract, an important feature that has also been reported in the SARS-CoV-2 infection. The outbreak of the pandemic required safety measures, including isolation that affected psychosocial health, with long-term repercussions even among children, where the issues seem to be more complex compared with adults [4,22,23,24].

It is uncertain if intestinal SARS-CoV-2 infection is secondary to respiratory infection or it is primarily, carried out via the oral-fecal route [5], a context in which it has been proposed that a pathogenic lung-intestine axis exists, based on the release of cytokines that bind to ACE-2 receptors present in both locations [25,26,27,28]. The hypothesis is even more plausible as viral infection also affects the nervous system, and together with the existence of the brain-intestine axis, the presence of FAPD seems to be justified [5,29].

In our study, we identified different distributions of FAPDs for the three groups. Thus, we found that most cases belonged to IBS, with a maximum prevalence of 91.3% in the pandemic positive group, followed by FD with 22.2% in the pre-pandemic group and AM with values of 18.2% in the pandemic negative group.

In this study, we used the Rome IV criteria which are unlikely in the case of organic disease [4]. Beyond clarifications that the Rome IV criteria support the FAPD diagnosis, there are limitations related to validity, since in the last two decades different methods of assessment of abdominal pain have been used through the world, almost 80% of them using the previous Rome III criteria. There are only few studies in the literature addressing functional abdominal pain compared in the pre-pandemic and pandemic COVID-19 period, and in which current diagnostic criteria are used. For example, Farello et al. analyzed the role of COVID-19 on FGIDs, and reported a higher number compared to pre-pandemic data, in terms of AM which almost tripled, and IBS which almost doubled [4]. This is consistent with our results showing the superiority of the two entities in the pandemic group. However, it should be mentioned that between the two studies there are differences related to the population analyzed and the clinical entities investigated.

There are other recent studies in which FAPDs have been investigated without reference to the COVID-19 pandemic [12,30]. In a meta-analysis of 58 articles using different methodologies, an overall prevalence of 8.8% for IBS, 4.5% for FD, and 1.5% for AM was reported [12]. In addition, population-based studies conducted on children and adolescents between 3–16 years of age and applying Rome III criteria showed a mean prevalence of 13.8 for IBS, with a range of 0–25.7% [10].

In terms of age and gender, there is massive heterogeneity in the results related to the presence of FAPD in the pediatric population, but indicating by a peak prevalence around the age of 11 years [10]. Studies that have included patients of ages close to those in our study are rare. Rask et al., investigating the functional somatic pain in 5–7-year-old children, showed a higher prevalence of these symptoms in females compared to males, 27.6 vs. 18.8% [31]. On a large group of 7-year-old children, Bakoula et al. indicate a prevalence of recurrent pain dependent on psychosocial conditions of 8.8% in females and of 5.7% in males [32]. In the meta-analysis conducted by Korterink et al., the prevalence of FAPD was higher in females, in 91.7% of cases [12]. The findings are similar to those obtained in our study in which females predominated in all groups analyzed.

Although easier access to the medical services for patients in urban areas, especially under restrictive measures, may influence the results, in this study, the urban/rural ratio was 1.42 in the pre-pandemic group, 0.83 in the COVID-19 negative pandemic group, and 10.5 in the COVID-19 positive pandemic group. The results even indicate an increase in the number of rural patients during the pandemic compared to the pre-pandemic period for children without viral infection, and the COVID-19 positive group is the dominant urban group, with an increase of almost 9-fold compared to the pre-pandemic group, and almost of10-fold compared to the COVID-19 negative pandemic group. The epidemiological studies confirm a viral spread and a higher incidence of COVID-19 from urban to rural areas, although morbidity and mortality rates may be significant in the latter due to poor socioeconomic status, and lack of medical compliance, and access to health services [33].

One important addition of the present study in evaluating FAPD was to determine the contribution that pandemic lockdown and viral infection had on the occurrence of this type of abdominal pain. The lockdown measures during the pandemic COVID-19 period had well documented negative overall psychological effects related to stress, fear of the unknown, depression, anxiety, and exacerbation of symptoms, effects that also extended to the pediatric population [4,20]. There are data indicating that pandemic stress is present and affects daily activity in many IBS patients, relative to the general population [20]. In this context, there is a major psychological component induced by the existence of the pandemic and social implications impacting the exacerbation of digestive symptoms. On the other hand, the presence of viral infection may further exacerbate these manifestations, although mortality and complication rates in children are low [34]. However, in our study, the results obtained indicate a different clinical-epidemiological FAPD profile for patients in the pre-pandemic period compared to the pandemic period, and with the COVID-19 positive group, which unlike the other groups was associated with a high number of IBS and urban patients, as well as a tendency to associate with males.

These data support the involvement of viral infection in the increased number of FAPD. There are reports of the occurrence of FAPD diagnosed according to Rome IV criteria as sequelae of SARS-CoV-2 infection, with the authors suggesting the possible etiological involvement of the virus [35]. On the contrary, there are studies indicating the absence of differences between the early pandemic period and similar pre-pandemic periods in terms of FGID or even a prevalence decreases during lockdown attributed to increased quality of life, reduced school-related stress and proximity to parents [36,37].

However, in this study, there are some differences in the COVID-19 negative pandemic group with at least double the number of urban IBS and FD compared to the pre-pandemic group, which could argue for favoring the pandemic situation of increasing the number of urban FAPD, but the statistical analysis did not confirm the results obtained. Nevertheless, the association is suggested in the recent study by Farello et al., which emphasizes the importance of combating lockdown-induced pandemic psychological stress to relieve symptoms of abdominal functional disorders [4].

The present study has some limitations, mainly related to the number of cases and time intervals. The study includes only a limited period of 6 months during the COVID-19 pandemic, which may not be representative of the entire period during which the viral infection occurred in Romania; however, the period for which the analysis was conducted is related to the higher number of cases diagnosed with FAPD in a certain time interval, which may also have been caused by a period of relaxation of restrictions after the first pandemic wave in this country. At the same time, the rather small number of cases was also influenced by the inclusion criteria of the study, including the limited age range of preschoolers, which we considered useful to obtain a homogeneous group to be analyzed. Another limitation of the study concerns possible subjective aspects related to exaggerated or altered descriptions of children’s pain, aspects that can generally be considered in studies examining functional pain. In our study, FAPD was assessed by a rigorous clinical-anamnestic examination that allowed the exclusion of the organic component.

Although the results of the study indicate an increase in the number of FAPD cases due to viral infection and suggests some influence of pandemic-induced psychosocial changes, studies on larger groups of patients are needed for confirmation, as well as on groups of school-aged children or adolescents, which may provide particular results.

Through the high rate of digestive manifestations, psychosocial burden and the risk of medical procedures in the COVID-19 era, we find the alteration of the management of digestive disorders in children [5], which led to an increase in the number of uninvestigated and undiagnosed cases. In the coming years there will be a major impact related to the clinical-epidemiological aspects in pediatric pathology, which requires the improvement of future monitoring and treatment strategies.

## 5. Conclusions

The study showed differences in the clinical-epidemiological profiles of FAPD associated with the pre-pandemic and pandemic COVID-19 period, as well as COVID-19 negative and positive patient groups. Viral infection has been associated with an increased number of IBS cases in urban areas, with a tendency to be diagnosed mainly in males. The results suggest the involvement of viral infection in the occurrence of FAPD given the underlying conditions of the pandemic, whose contribution remains to be clarified in future broader studies.

## Figures and Tables

**Figure 1 life-12-00509-f001:**
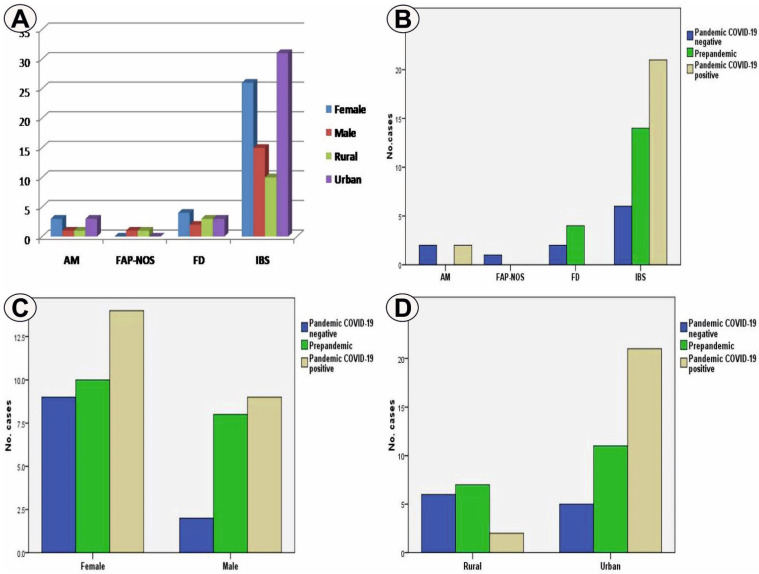
(**A**) Cases distribution in the entire study group depending on the analyzed parameters; (**B**) FAPD types distribution in the analyzed groups; (**C**) Gender distribution in the analyzed groups; (**D**) Environment origin distribution in the analyzed groups.

**Figure 2 life-12-00509-f002:**
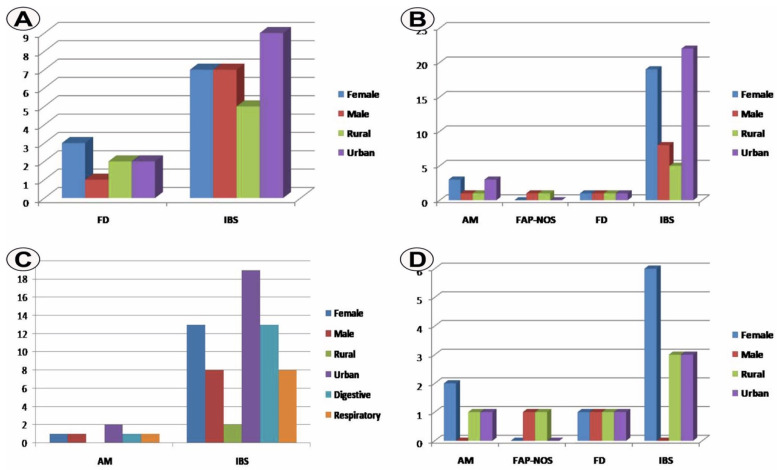
(**A**) Cases distribution in the pre-pandemic group depending on the analyzed parameters; (**B**) Cases distribution in the pandemic group depending on the analyzed parameters; (**C**) Cases distribution in the pandemic COVID-19 positive group depending on the analyzed parameters; (**D**) Cases distribution in the pandemic COVID-19 negative group depending on the analyzed parameters.

**Table 1 life-12-00509-t001:** Rome IV criteria for the diagnosis of functional abdominal pain disorders [21].

**IBS**	symptoms must persist for at least 2 months and include: abdominal pain at least 4 days/month associated with changes in stool frequency and appearance; in children with constipation, the remission of constipation is not accompanied by the remission of pain; symptoms cannot be explained by another condition.
**FD**	one or more of the following symptoms persist for at least 2 months, present at least 4 times/month: postprandial fullness, early satiety, epigastric pain or burning not associated with defecation; symptoms cannot be fully explained by another medical condition.
**AM**	mandatory presence of the following symptoms manifested at least 2 times within 6 months: paroxysmal episodes of intense, acute, periumbilical, median, or diffuse abdominal pain lasting 1 hour or more; episodes are separated by long intervals of time (weeks to months); pain is disabling and interferes with normal activities; pain is associated with at least 2 or more of the following: anorexia, nausea, vomiting, headache, photophobia, pallor; symptoms cannot be fully explained by another medical condition.
**FAP-NOS**	presence of symptoms that persist for at least 2 months, at least 4 times/month represented by: episodic or continuous abdominal pain that does not occur only during physiological events (i.e., eating, menstruation); insufficient criteria for irritable bowel syndrome, functional dyspepsia or abdominal migraine; symptoms cannot be fully explained by another medical condition.

**Table 2 life-12-00509-t002:** Cases distribution according to the analyzed parameters.

Parameters/No. Cases (%)		COVID-19 Negative (Prepandemic) **	COVID-19 Negative (Pandemic) **	COVID-19 Positive (Pandemic) **	Total *
FAPD	IBS	14 (77.8%)	6 (54.5%)	21 (91.3%)	41 (78.8%)
	FD	4 (22.2%)	2 (18.2%)	0	6 (11.5%)
	AM	0	2 (18.2%)	2 (8.7%)	4 (7.7%)
	FAP-NOS	0	1 (9.1%)	0	1 (2%)
Gender	Female	10 (55.6%)	9 (81.8%)	14 (60.9%)	33 (63.5%)
	Male	8 (44.4%)	2 (18.2%)	9 (39.1%)	19 (36.5%)
Environment	Urban	11 (61.1%)	5 (45.5%)	21 (91.3%)	37 (71.2%)
	Rural	7 (3.9%)	6 (54.5%)	2 (8.7%)	15 (28.8%)
Viral symptoms	Digestive	-	-	14 (60.9%)	14 (26.9%)
	Respiratory	-	-	9 (39.1%)	9 (17.3%)
Total *		18 (34.6%)	11 (21.2%)	23 (44.2%)	52

Note: * reported to the entire study group; ** reported to separated study groups.

## Data Availability

The data presented in this study are available on request from the corresponding author.

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
