# Peer review of "Pediatric Functional Abdominal Pain Disorders following COVID-19"

_life, 2022, doi:10.3390/life12040509_

Round 1
Reviewer 1 Report
Thank you for giving me the chance to review the revised resubmission of this paper.
It was deeply ameliorated. However, I have a few concerns that should be addressed.
Figure 1: I would put different column groups (AM, FAP, IBD,...) according to pre-pandemic, covid negative and covid positive.
Discussion must be shortened.
English must be improved as too many grammar errors.
Author Response
Februrary 22, 2022
Re: Revision of Manuscript life-1606991
Dear Academic Editor of the Life Journal,
Thank you and the reviewers for the careful consideration of our manuscript and informative comments. Please consider the present letter along with the uploaded manuscript.
Please let us know if we can address any further comments for our manuscript to fully meet your publication criteria.
Best wishes,
Corina Vasile, MD
Corresponding author
Response to reviewer’s comments:
Reviewer 1
Dear respected reviewer,
Thank you one more time for the constructive comments. Following your suggestions, we have made some changes to improve the editing process and tracking of information. These changes refer to the following issues:
1.Reviewer 1: Figure 1: I would put different column groups (AM, FAP, IBD,…) according to pre-pandemic, covid negative and covid positive.
Author’s response:
We have modified Figure 1, to better illustrate the analyzed parameters, not only in terms of FAPDs but also related to gender and environment.
2.Reviewer 1: Discussion must be shortened.
Author’s response:
Discussions were shortened, with the concentration or reformulation of information of interest and the elimination of those that were repeated in the Methodology or Introduction.
3.Reviewer 1: English must be improved as too many grammar errors.
Author’s response:
We performed a thorough English language revision and corrected the grammar errors using Trinka AI.
Additional comment
We hope that these arguments/clarifications will plead for a study that is intended to be objective both by methodology and by the results reported to the literature data, even if these results can indicate only the clinical-epidemiological trends of the analyzed parameters, trends that need to be further investigated, and which may underlie future strategies for investigation and treatment in clinical practice.
Thank you once again!
Kind regards!

Reviewer 2 Report
The authors investigated viral-mediated sensitivity of the gastrointestinal tract in preschool children in the context of the COVID-19 pandemic in this work. Although the research means and perspectives are interesting, this manuscript may be unsuitable for publication in its current state. Since the number of cases was only 52, it might be arbitrary and hasty to conclude that viral infection was predominantly associated with an increased number of IBS diagnosed among males in urban areas.
Author Response
Februrary 22, 2022
Re: Revision of Manuscript life-1606991
Dear Academic Editor of the Life Journal,
Thank you and the reviewers for the careful consideration of our manuscript and informative comments. Please consider the present letter along with the uploaded manuscript.
Please let us know if we can address any further comments for our manuscript to fully meet your publication criteria.
Best wishes,
Corina Vasile, MD
Corresponding author
Response to reviewer’s comments:
Reviewer 2
Dear respected reviewer,
Firstly, we want to thank you for the comments, which especially appreciated the investigated topic and the perspectives it can open.
1.Reviwer 2: The authors investigated viral-mediated sensitivity of the gastrointestinal tract in preschool children in the context of the COVID-19 pandemic in this work. Although the research means and perspectives are interesting, this manuscript may be unsuitable for publication in its current state. Since the number of cases was only 52, it might be arbitrary and hasty to conclude that viral infection was predominantly associated with an increased number of IBS diagnosed among males in urban areas.
Author’s response:
We would like to mention some aspects related to the group analyzed in this study:
- For this investigation, a major concern was to obtain a study group as homogeneous as possible, diagnosed based on anamnestic and clinical objective criteria. We believe this is important in the context in which the results of studies in the literature related to FAPD are heterogeneous, these being performed on different age ranges, with the use of variable diagnostic criteria and a variable methodology,
-Thus, the pre-pandemic and pandemic groups were subjected to inclusion and exclusion criteria that may explain the rather small number of cases. In addition to these criteria, for pre-pandemic cases, we opted for an equivalent interval in terms of duration and location in time, in order to reduce the influence of social and environmental factors on the results,
-Regarding the analyzed pandemic period, it coincided in Romania with a period of relative relaxation of the imposed restrictions, which stimulated the medical presentation of the patients with FAPD with/without COVID-19 infection; In this sense, during the analyzed period, we identified the highest number of FAPDs,
-Of course, although the study methodology can explain the number of cases, overall it remains a reduced one, being necessary studies on larger groups in number and as pediatric age ranges investigated. For this reason, we have introduced a section with the limits of the study, which gives a broad indication of future perspectives in relation to these results.
Additional comment
We hope that these arguments/clarifications will plead for a study that is intended to be objective both by methodology and by the results reported to the literature data, even if these results can indicate only the clinical-epidemiological trends of the analyzed parameters, trends that need to be further investigated, and which may underlie future strategies for investigation and treatment in clinical practice.
Thank you once again!
Kind regards!

Round 2
Reviewer 2 Report
Thanks for your response.
Author Response
Dear Editor
In order to make it easier to follow the latest changes made, in the new submitted manuscript, the old Track Changes were accepted.
Thank you!
-----------------------------------------------------------------------------------------------------------
Dear Respected Editor,
Firstly we want to thank you for the constructive comments and clear requirements, which helped us to improve our study and to correct the errors. We hope the new submitted material will bring more accuracy and conciseness in the information provided. The changes made refer to the following issues:
Q1 (question): Firstly, a major language revision is necessary since some passages are hard to be understood (see i.e. par. Exposure Assessment line 148).
R1 (response): As suggested by the Referee, we have now made a major language revision and rephrased some sentences (including the paragraph in question) so that the information is clearer and better understood.
Q2: Secondly, it is not clear how, and with which rationale, the subjects were divided into groups. Authors stated that the group "pandemic" was "exposed to COVID-19 infection" (maybe you meant "SARS-COV-2 infection") and that "in the pandemic group, only patients with exposure to the viral infection no more than 6 moths ago were included". So why authors divided between COVID-19 positive and negative? Does that refer to whether subjects tested positive for sars-cov-2 had or hadn't developed the COVID-19 disease? This is a crucial aspect for interpreting the results.
R2_1: In accordance with the error reported by the Referee, and as in fact, it is specified in the first sentence of the Introduction section, about the notions of COVID-19 and SARS-CoV-2, in the whole manuscript the wrong term of "COVID-19 infection" has been replaced by "SARS-CoV-2 infection" or "viral infection".
R2_2: In this study we considered that, in addition to the potential information related to the distribution of FAPDs and the associated epidemiological profile in pandemic and prepandemic groups, the stratification of pandemic patients into positive and negative cases may clarify the contribution of the pandemic context versus the viral infection itself in the variation of cases distribution. The premise was established so that the differences between the positive versus negative pandemic groups could be attributed to the viral infection and to the differences between the negative prepandemic versus negative pandemic groups could be attributed to the pandemic context.
For the pandemic group that was exposed to the viral infection, positive cases for SARS-CoV-2 infection were confirmed by RT-PCR testing, thus being included in the Covid-19 positive pandemic group. For the same pandemic group, cases with FAPD-specific digestive symptoms, which were negative for SARS-CoV-2 infection by RT-PCR tests and had no history of medical visits or other positive tests after inclusion in the study, were included in the Covid-19 negative pandemic group. This issue has now been clarified in section 2.4: Exposure assessment.
Q3: Lastly, the analysis is merely descriptive and this hampers any conclusions. A rigorous statistical analysis on the difference of incidence between groups, and weight of covariates, is needed before this MS could be endorsed for publication.
R3: Initially, the analyzed group included only cases from the pandemic period (34 cases), being accompanied by the statistical analysis performed by comparative tests of the cases distribution. Because the number of cases was limited, the reviewers recommended removing the statistical analysis and conducting a descriptive analysis that would have been more useful in medical practice. Subsequently, the prepandemic group was identified and introduced into the study (18 cases), for which we continued the comparative descriptive analysis.
Following the requirement, the descriptive analysis has been maintained, and we reintroduced the statistical analysis, using case distribution comparison tests, in relation to the analyzed parameters, respectively chi-squared test (χ2) within the SPSS software, already used for graphical representation and in the table.
We chose this type of statistical analysis because the analyzed groups are still rather small (one of the limitations of the study which is in fact mentioned in the body text of the paper), the selection of cases was not randomized and we do not have a reference population in terms of FAPD distribution or gender and associated environment that would allow the adjustment of weights. The absence of a reference population is due to the constantly improvement of the diagnostic criteria and the regional particularities of the distribution of different forms of FAPD, but also to the limited age range investigated (other study limit included in text of the paper).
Please note that the statistical analysis brought some changes to the final interpretation of the results that were included in the manuscript. In order to be able to follow up the text changes, we have resubmitted the manuscript utilizing the Track Changes feature so that the modifications are more clear.
Thank you once again!
Best wishes!

This manuscript is a resubmission of an earlier submission. The following is a list of the peer review reports and author responses from that submission.
Round 1
Reviewer 1 Report
Although the topic is interesting, the paper is very disorganized and very difficult to follow.
Inclusion criteria are very confusing.
Methods: was any control population analyzed? How were FAPD defined?
Statistical analysis: the sample size does not allow any analysis. A descriptive study would be more effective.
Also, a comparison with a comparable population in the pre-covid period would be an added value. May the symptoms be related also to the pandemic situation rather than to the infection?
Reviewer 2 Report
Respected authors,
I have reviewed your work, and I feel that the topic of your work is very novel, but the presentation of the work is not clear. I am writing some of my comments for improving the existing quality of your work.
Abstract:
The abstract is written very nicely. I feel that there are many unnecessary abbreviations, which are not defined scientifically. MDPI journal has no word count limit, I suggest removing all the unnecessary abbreviations because it is causing trouble to read.
Moreover, in the result section of abstract line number 33-36. I suggest you at first discuss the prevalence of various types of FAPD in one sentence. The following sentence presents the high-risk groups, i.e., gender, urban residence, and pathological history.
The index case of COVID-19 was reported from Wuhan city, China in December 2019, so it is not possible that the investigation was carried out before the disease index case. I suggest you correct the investigation date from abstract line number 29.
Introduction:
The start of the introduction section is very abrupt, I think at first you need to define what is coronavirus? What is the causative agent for coronavirus? What are the sign and symptoms? Then connect the topic sentence with the first sentence of your writeup.
I feel that the introduction section is very lengthy, and I did not observe coherence and cohesion between different paragraphs of the introduction section. For creating coherence and cohesion between different paragraphs, you need to connect the sentences and paragraphs with each other.
Certain paragraphs of the introduction section, i.e., from paragraphs 4 to 6 are not relevant to the study topic. These paragraphs described the factors for the low prevalence of COVID in children. However, the objective of this study was to assess Gastrointestinal diseases in COVID-positive children. I suggest you delete these paragraphs. Rather add some new paragraphs, which explicitly describe the gastrointestinal symptoms of COVID and why it is necessary to this study.
Methodology
I feel that you need to write your methodology in detail. You need to add some sub-headings in your methodology section. I am proposing a few headings: a) study setting and study design, b) study population and their eligibility criteria, c) sample size and sampling methods, d) exposure assessment, e) outcome assessment, f) study covariates, g) statistics, and h) ethical principles
There is a need to describe ROME IV criteria.
Why have you excluded children aged below four years? Was there any specific reason? If you have data then you can include them in your analysis.
In the outcome assessment method, clearly define all the terminologies related to FGID, and also write what methods were used for the assessment of each outcome.
Results:
The sample size of this study is not enough, so I suggest you remove inferential statistics. Rather present the findings descriptively. I feel that the descriptive findings of this study will also open avenues for clinicians to examine gastrointestinal health in children with COVID infection.
For results presentation create two columns: one for COVID positive and one for COVID negative, and write frequency and percentages of each outcome and other covariates in those columns.
Discussion:
The discussion is very lengthy and needs to be summarized. There is a need to discuss the study findings with other relevant literature.
There is a need to add a limitation section.